

# Identification of Vitamin D-related gene signature to predict colorectal cancer prognosis

Luping Bu[1,2,*], Fengxing Huang[1,2,*], Mengting Li[1,2], Yanan Peng[1,2], Haizhou Wang[1,2], Meng Zhang[1,2], Liqun Peng[1,2], Lan Liu[1,2] and Qiu Zhao[1,2]

[1] Hubei Clinical Center and Key Lab of Intestinal and Colorectal Diseases, Wuhan, China
[2] Department of Gastroenterology, Zhongnan Hospital of Wuhan University, Wuhan, China
* These authors contributed equally to this work.

Corresponding authors
Lan Liu, lliugi@whu.edu.cn
Qiu Zhao, qiuzhao@whu.edu.cn

## ABSTRACT

Colorectal cancer (CRC) is one of the most common malignant carcinomas worldwide with poor prognosis, imposing an increasingly heavy burden on patients. Previous experiments and epidemiological studies have shown that vitamin D and vitamin D-related genes play a vital role in CRC. Therefore, we aimed to construct a vitamin D-related gene signature to predict prognosis in CRC. The CRC data from The Cancer Genome Atlas (TCGA) was performed as the training set. A total of 173 vitamin D-related genes in the TCGA CRC dataset were screened, and 17 genes associated with CRC prognosis were identified from them. Then, a vitamin D-related gene signature consisting of those 17 genes was established by univariate and multivariate Cox analyses. Moreover, four external datasets (GSE17536, GSE103479, GSE39582, and GSE17537) were used as testing set to validate the stability of this signature. The high-risk group presented a significantly poorer overall survival than low-risk group in both of training set and testing sets. Besides, the areas under the curve (AUCs) for signature on OS in training set at 1, 3, and 5 years were 0.710, 0.708, 0.710 respectively. The AUCs of the ROC curve in GSE17536 for 1, 3, and 5 years were 0.649, 0.654, and 0.694. These results indicated the vitamin D-related gene signature model could effectively predict the survival status of CRC patients. This vitamin D-related gene signature was also correlated with TNM stage in CRC clinical parameters, and the higher risk score from this model was companied with higher clinical stage. Furthermore, the high accuracy of this prognostic signature was validated and confirmed by nomogram model.
In conclusion, we have proposed a novel vitamin D-related gene model to predict the prognosis of CRC, which will help provide new therapeutic targets and act as potential prognostic biomarkers for CRC.

## INTRODUCTION

Colorectal cancer (CRC) remains the third most common malignant tumor worldwide and is the second one in cancer-related deaths (*Ferlay et al., 2018*). In China, CRC is the third

one in annual incidence and also the leading cause of tumor-related deaths (*Zhang et al., 2019*). The overall survival of CRC is unsatisfactory, and the five-year overall survival rate is just over 50% (*Frampton & Houlston, 2017*). CRC is highly malignant and causes a tremendous economic burden on patients (*Luengo-Fernandez et al., 2013*). Therefore, the poor prognosis and the increasing incidence of CRC have provided strong motivation to construct a predictive model in CRC patients, which will benefit personalized treatment in clinical management.

Vitamin D is a fat-soluble vitamin and there are many genes related to its metabolism and action (*Fedirko et al., 2019*). Vitamin D can be obtained from diet or the endogenous synthesis in the epidermis under sunlight exposure (*Saraff & Shaw, 2016*). It has been demonstrated that vitamin D benefits clinical outcome and improves the long-term survival of CRC patients (*Xu et al., 2021*). Besides, circulating vitamin D may be a CRC biomarker and vitamin D deficiency is closely related to the high incidence of CRC (*He et al., 2018*; *Meeker et al., 2016*; *Wu et al., 2020*). Vitamin D has been reported to inversely correlated with CRC and has a protective effect on CRC in clinical studies (*Jenab et al., 2010*; *Manousaki & Richards, 2017*; *Meeker et al., 2016*). Mechanically, vitamin D inhibits tumor growth and affects CRC development in animal models (*Hummel et al., 2013*; *Newmark et al., 2009*). Besides, it has been reported that vitamin D inhibits proliferation and promotes differentiation in CRC cells (*Leyssens et al., 2015*). Many genes are related to vitamin D metabolism and action, play an essential role in tumors; for example, CYP24A1, an important vitamin D-related gene, was up-regulated in CRC patients and nominated as a promising biomarker (*Sadeghi & Heiat, 2020*). Vitamin D and its related genes are correlated with the homeostasis of the intestinal epithelium, regulate immune cells, and may protect against colon cancer (*Cantorna, Snyder & Arora, 2019*). In a word, vitamin D and its related genes were closely correlated with CRC tumor pathogenesis.

Prognostic model based on gene set, with a higher predictive value than a single gene, was constructed in previous studies to predict the clinical outcome of patients. However, it has not been explored whether vitamin D-related gene signature could be biomarkers for the prognosis of CRC. A systematic functional study of vitamin D-related genes in CRC will contribute to our more profound understanding of vitamin D-related genes and provide new ideas for the pathogenesis of CRC. We aimed to clarify whether vitamin D-related genes can help CRC patients predict the prognosis and provide a better diagnosis and treatment. Therefore, we focused on constructing a prognostic signature based on vitamin D-related genes.

## MATERIALS AND METHODS

### Data collection

The studies involving human participants were reviewed and approved by the Medical ethics committee of Zhongnan Hospital of Wuhan University. Gene expression matrix data of CRC samples were downloaded from The Cancer Genome Atlas (TCGA) data portal (https://portal.gdc.cancer.gov) and the Gene Expression Omnibus (GSE17536, GSE103479, GSE39582, and GSE17537) data pool (GEO, https://www.ncbi.nlm.nih.gov/geo/)

as training and external validation sets, respectively (up to December 16, 2020) (*Sun et al., 2019*). Transcriptome information for CRC was included. These were the inclusion criteria in this study: (1) The samples with both mRNA sequencing data and survival status from the CRC patients; (2) The samples with complete clinical information. The expression data of TCGA of 480 CRC and 41 non-tumorous tissues was the HT Seq-FPKM type and has been normalized. In TCGA database, there were 453 CRC patients with overall survival status, the expression matrix was arranged with "dplyr" package (*Günalan, Cebioğlu & Çonak, 2021*) and "data.table" (*Zhou et al., 2020*) package in R. Then, we searched the clinical information of the patients from the database, including survival information, age, gender, tumor stage, and clinical classification; 393 CRC patients with complete clinical information were selected for further analysis. And GEO databases (GSE17536, GSE103479, GSE39582, and GSE17537), containing 177, 155, 562, and 55 CRC patients, were normalized using "limma" package in R. A total of 194 vitamin D–related gene lists were obtained from two literature pieces (*Fedirko et al., 2019*; *Pytel et al., 2019*). Among them, 173 vitamin D–related genes were identified from TCGA.

## Construction and validation of the prognostic vitamin D-related gene signature for CRC

The 453 CRC samples from TCGA were downloaded as training set. To establish the prognostic model, univariate and multivariate Cox analysis was applied to examine the predictive value of vitamin D-related genes in CRC patients with overall survival by using the "survival" package in R (*Tang et al., 2017*). A $P$-values of $< 0.05$ were considered significant and shown by a forest plot. After acquiring the prognostic vitamin D-related genes of CRC, genes with a hazard ratio (HR) > 1 were identified as high-risk genes, while HR < 1 was identified as low risk. The model was utilized to evaluate the correlation between overall survival (OS) and vitamin D-related genes. Finally, risk scores were calculated based on the corresponding Cox regression coefficient and the expression levels of 17 genes of 453 CRC patients in TCGA, and the formula was shown as risk score = expression of vitamin D-related gene 1 × coefficient 1 + expression of vitamin D-related gene 2 × coefficient 2 + vitamin D-related gene n × coefficient n. Patients were divided into high and low-risk groups based on the median risk score. Kaplan–Meier (KM) analysis was used to compare survival between high-risk and low-risk groups with "survival" package and "survminer" package in R. To verify the stability of the signature, the receiver operating characteristic (ROC) analysis was performed, and the AUC was calculated by the R package "survivalROC" (*Wang et al., 2018*). Then, we downloaded four datasets (GSE17536, GSE103479, GSE39582, and GSE17537) in the Gene Expression Omnibus database (GEO database) to validate the model.

## Independent prognostic analysis and clinical correlation analysis

The clinical information was selected including age, gender, clinical stage, tumor-node-metastasis (TNM) status. To evaluate whether the risk score is an independent prognostic factor associating with overall survival, univariate and multivariate independent prognostic analyses were performed with the "survival" package. A $p$-value of $< 0.05$ was

considered significant statistically. Then, time-dependent ROC curve was analyzed, and the area under the curve (AUC) was calculated using the "survivalROC" package. A clinical relevance analysis was conducted using the "beeswarm" package (*Bi et al., 2019*) to assess the correlation between prognostic vitamin D-related genes and clinical information. All results were considered to be statistically significant at $p < 0.05$.

## Nomogram model construction

A nomogram based on the results of previous multivariate Cox regression analyses was constructed by using the R language "rms" package and "hmisc" package (*Jiang et al., 2019*; *Wells et al., 2018*). Calibration curve were generated to analyze the agreement between nomogram and ideal observation. By quantifying the net benefits under different threshold probabilities, a decision curve analysis is performed to evaluate the clinical utility of the predictive nomogram.

## Statistical analysis

Statistical analysis was made using R software (*Chan, 2018*), with version number v4.0.3. Kaplan–Meier analysis was applied to estimate the overall survival rate of different groups, and log-rank was used to test the significance of the difference between other groups. The plot and the heatmap were generated using the R package "ggplot2" (*Günalan, Cebioğlu & Çonak, 2021*) and "pheatmap" (*Zhang et al., 2020*).

# RESULTS

## Identification of prognostic vitamin D-related genes in CRC

The workflow of this study was illustrated in Fig. 1. A univariate Cox analysis was performed on vitamin D-related genes in 453 CRC samples, and 17 genes related to overall survival were identified from 173 vitamin D–related genes (Fig. 2, Table 1). Among them, 14 genes (CYP24A1, TGFB1, IGFBP2, IGFBP3, VGF, AGAP2, DENND6B, LRRC8A, BCL6, FCER2, ELL, CD36, AGPAT1, and TMCO6) were ranked as high-risk genes (HR > 1), while 3(NCOA7, GSR, and MPC1) were low-risk genes (HR < 1) according to the HR.

## Construction and validation of vitamin D-related prognostic model for CRC

The complete clinical information of 393 CRC patients was collected including survival status, survival time, age, gender, clinical stage, and TNM stage. A prognostic model was developed based on above identified 17 genes by performing multivariate Cox analyses (Table 1). The CRC patients were divided into a high-risk group ($n = 226$) and a low-risk group ($n = 227$) according to the risk score's median cutoff (Fig. 3A). The ranked risk scores of patients and their survival status in the training set and the testing sets were plotted, respectively. Riskscore = (0.237883456*CYP24A1) +(−0.014875163*NCOA7)+(−0.000370145*TGFB1)+(−0.019371671*GSR) +(0.005620769*IGFBP2)+(−0.0000458*IGFBP3)+(0.004647164*VGF) +(0.148455838*AGAP2)+(0.406871954*DENND6B)+(0.011891263*LRRC8A)
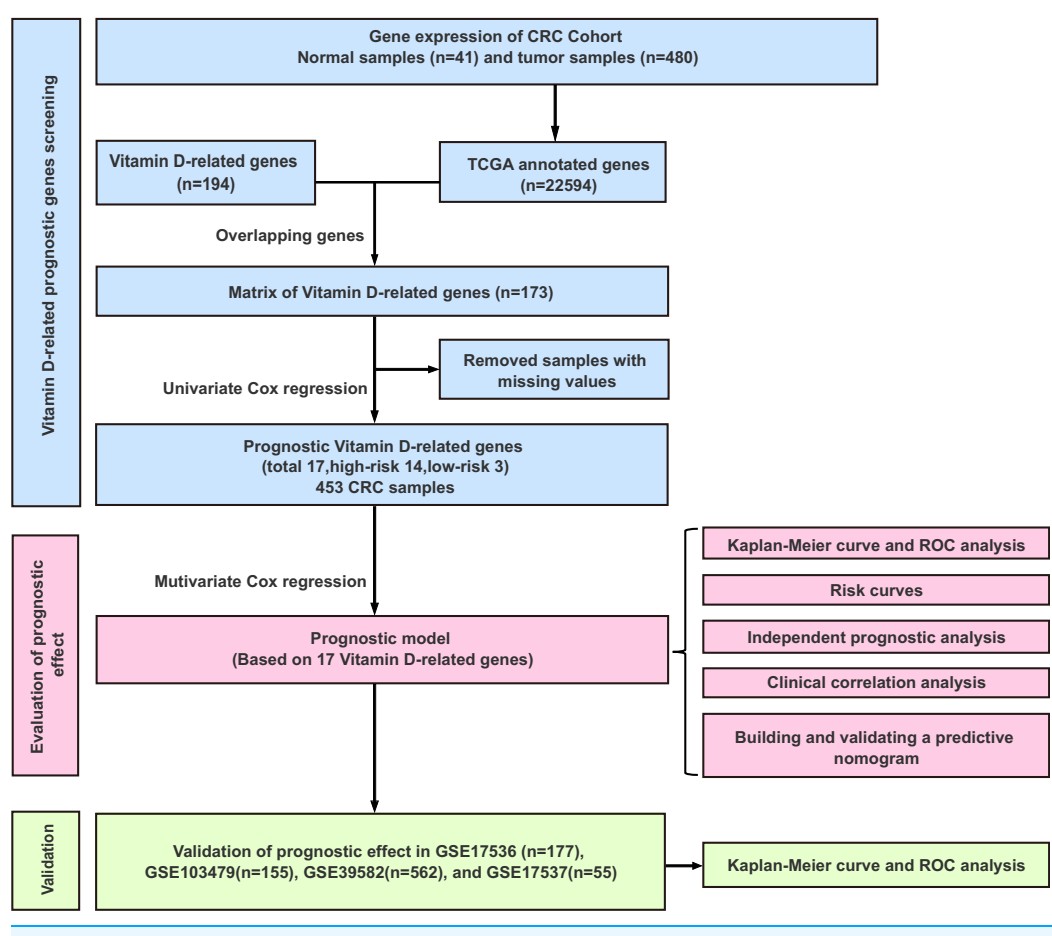

**Figure 1 Experimental flowchart for construction and validation of a 17 vitamin D-related gene prognostic model.**

+(−0.045790322*BCL6)+(0.048350529*FCER2)+(0.029666421*ELL) +(0.000270836*MPC1)+(0.055492489*CD36)+(0.015772143*AGPAT1) +(0.072341304*TMCO6). A survival status overview was established (Fig. 3A, Figs. 4A, 4C, 4E, 4G). The results represented that patients in the high-risk group showed a higher mortality rate than those in a low-risk group in both of training set and testing sets.

## Predictive performance of the vitamin D-related gene signature

Kaplan–Meier survival analysis was performed between high- and low-risk groups in both of training set and testing sets to investigate the prognostic value of the vitamin D-related gene risk signature. The high-risk group had a significantly poorer OS than that of the low-risk group in the training set (Fig. 3A) and the testing sets (Figs. 4A, 4C, 4E, 4G). The ROC analysis was performed to evaluate the predictive efficiency of the vitamin D-related gene signature. The AUCs for the signature on OS at 1, 3, and 5 years were 0.710, 0.708, 0.710 in the training set (Fig. 3B). The AUCs for the signature on OS at 1, 3, and 5 years were 0.649, 0.654, and 0.694 in the GSE17536 (Fig. 4B), 0.676, 0.600, 0.588 in GSE103479 (Fig. 4D), 0.595, 0.675, 0.672 in GSE39582 (Fig. 4F), and 0.896, 0.923, 0.861 in GSE17537 (Fig. 4H). These results showed that the model could predict the survival status

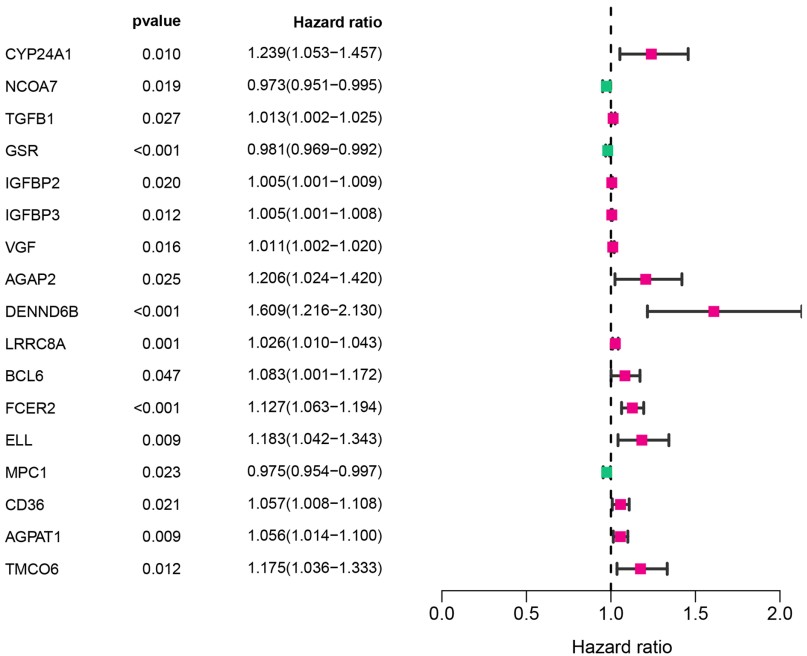

**Figure 2 Assessment of vitamin D-related genes in predicting prognosis of CRC exhibited by forest plot.**

| Gene symbol | HR | HR(95% CI) | P-value |
|---|---|---|---|
| CYP24A1 | 1.24 | [1.05–1.46] | 0.009832 |
| NCOA7 | 0.97 | [0.95–1.00] | 0.018687 |
| TGFB1 | 1.01 | [1.00–1.03] | 0.026872 |
| GSR | 0.98 | [0.97–0.99] | 0.000786 |
| IGFBP2 | 1.00 | [1.00–1.01] | 0.020022 |
| IGFBP3 | 1.00 | [1.00–1.01] | 0.01205 |
| VGF | 1.01 | [1.00–1.02] | 0.015677 |
| AGAP2 | 1.21 | [1.02–1.42] | 0.024878 |
| DENND6B | 1.61 | [1.22–2.13] | 0.00088 |
| LRRC8A | 1.03 | [1.01–1.04] | 0.00146 |
| BCL6 | 1.08 | [1.00–1.17] | 0.04722 |
| APBB3 | 1.22 | [1.04–1.42] | 0.012596 |
| FCER2 | 1.13 | [1.06–1.19] | 6.34E-05 |
| ELL | 1.18 | [1.04–1.34] | 0.00942 |
| MPC1 | 0.97 | [0.95–1.00] | 0.023354 |
| CD36 | 1.06 | [1.01–1.11] | 0.020768 |
| AGPAT1 | 1.06 | [1.01–1.10] | 0.008734 |
| TMCO6 | 1.18 | [1.04–1.33] | 0.011792 |

**Table 1 Seventeen vitamin D-related genes in the prognostic model of CRC.**

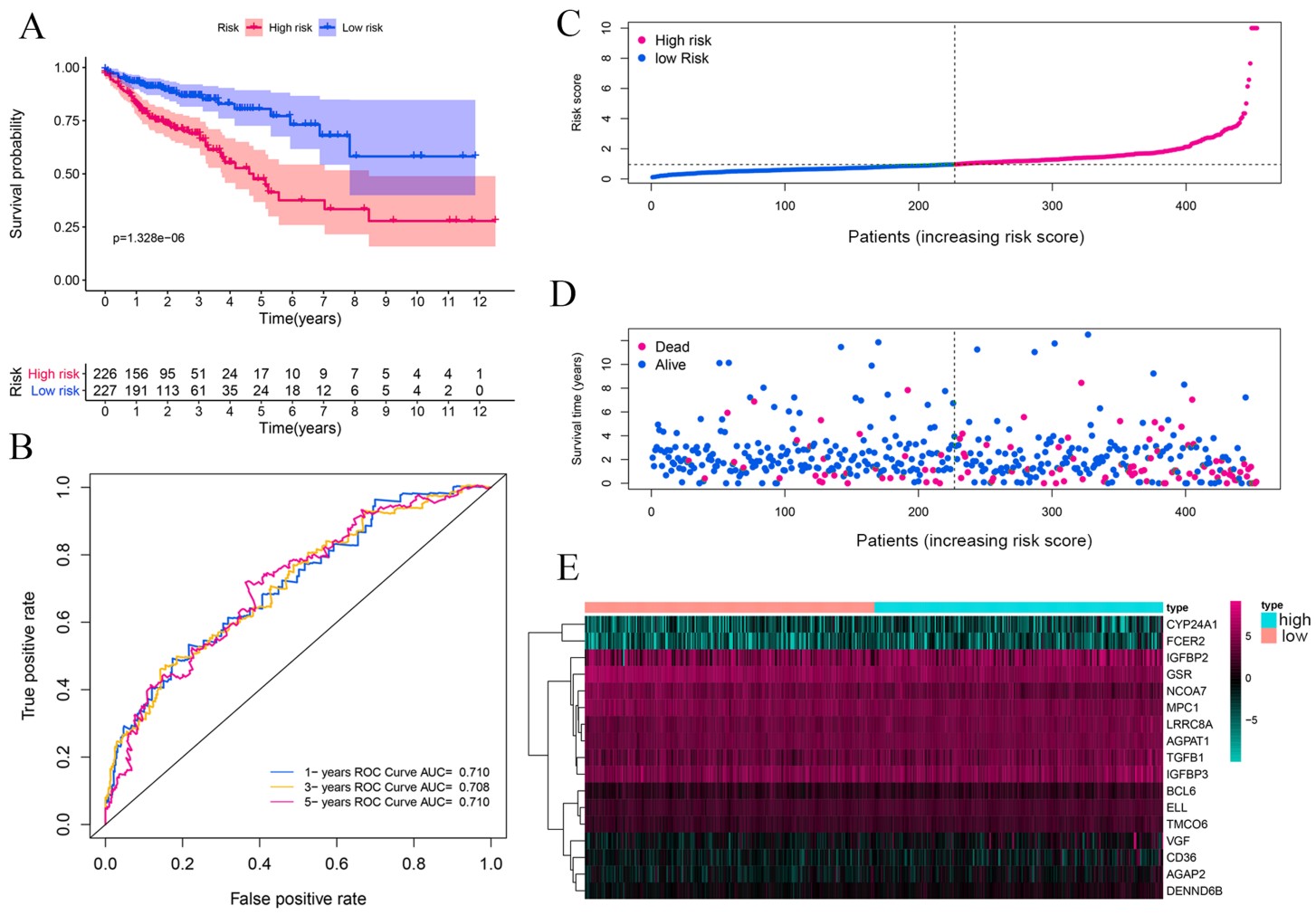

**Figure 3 Vitamin D-related prognosis genes are significantly correlated with the overall survival of CRC in the training set.** Kaplan–Meier analysis between high- and low-risk groups of TCGA CRC patients, the overall prognosis of CRC patients with high risk score is poor (A). ROC analysis of the risk score to assess the sensitivity and specificity (B). The relationship between risk score, death, and expression of characteristic genes (C–E).

of CRC patients with high accuracy, indicating good sensitivity and specificity. Detailed clinical characteristics of the TCGA CRC patients were listed in Table 2. The results suggested that the differences in the distribution of the clinical stage and TNM classification in the two groups were statistically compelling (Table 2). Besides, our model was proved to be an independent factor for estimating the prognosis of CRC (Figs. 5A, 5B). The univariate analysis displayed that age ($P = 0.002$), pathological stage ($P < 0.001$), TNM classification ($P < 0.001$), and the vitamin D-related prognostic model ($P < 0.001$) were significantly correlated with overall survival. The multivariate analysis indicated that vitamin D-related prognostic models ($P < 0.001$), age ($P < 0.001$), and T classification ($P = 0.046$) remained independent prognostic factors related to OS (Table 3). Overall, vitamin D-related prognostic model could effectively predict the survival of CRC patients.

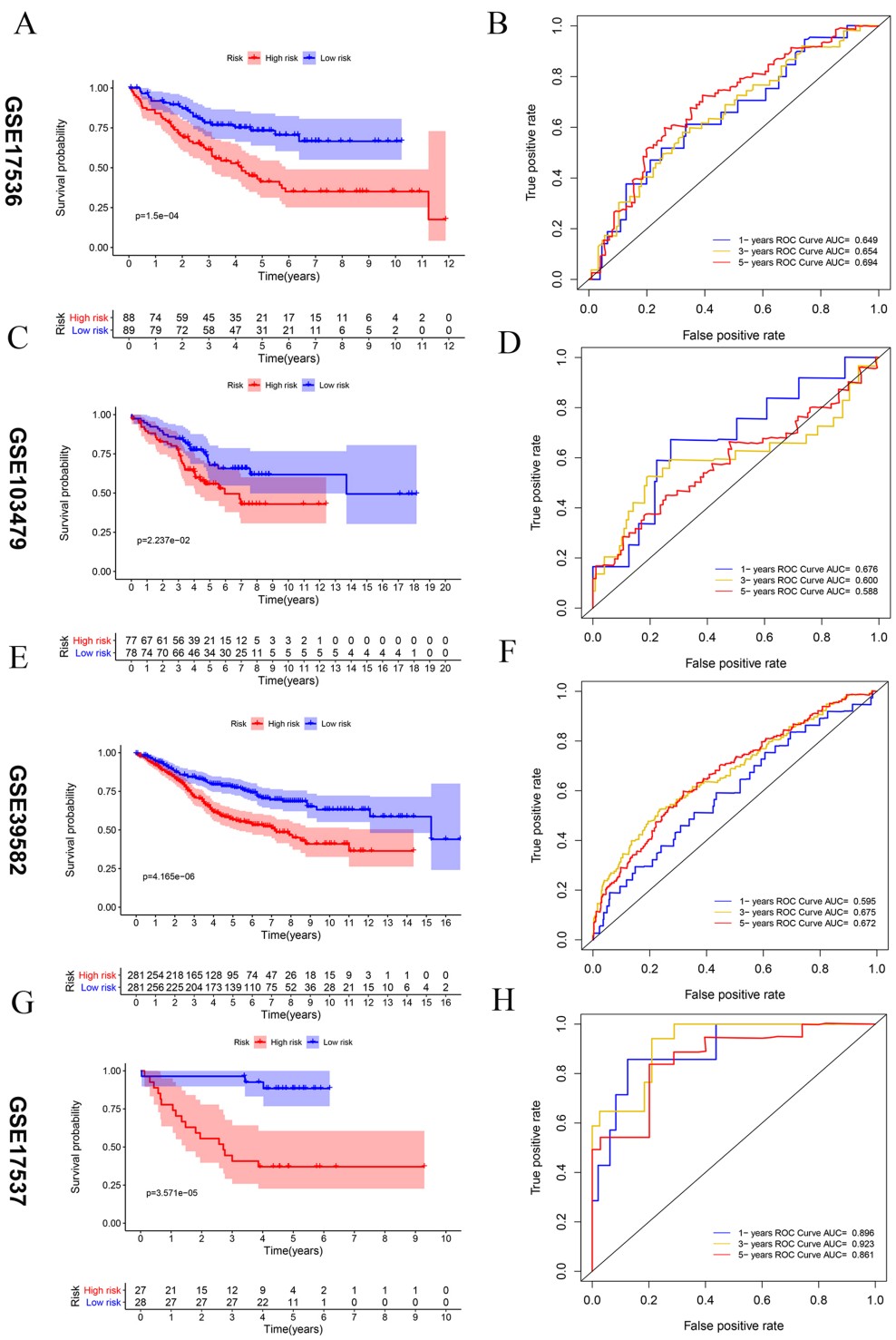

**Figure 4 Vitamin D-related prognosis genes are significantly correlated with the overall survival of CRC in four testing sets.** Four external datasets (GSE17536, GSE103479, GSE39582, and GSE17537) were used as testing set to validate the stability of this signature. Kaplan-Meier analysis between high- and low-risk groups of GEO database CRC patients, the overall prognosis of CRC patients with high risk score is poor (A, C, E, and G). ROC analysis of the risk score to assess the sensitivity and specificity (B, D, F and H).

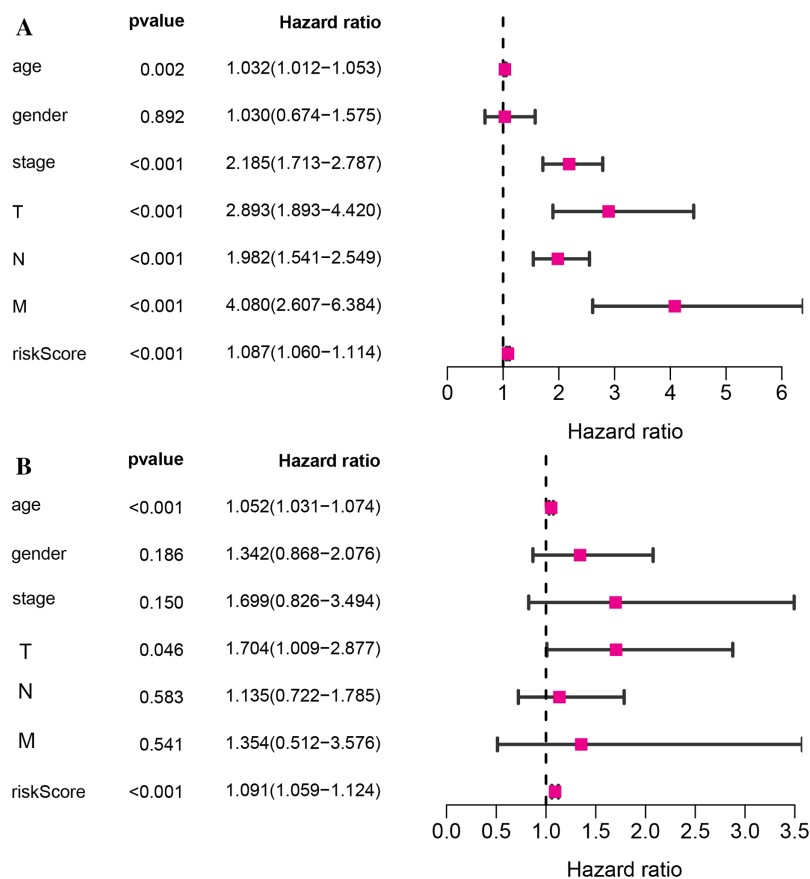

**Figure 5** **The relationship of the vitamin D-related signature and clinical factors with overall survival in TCGA dataset.** Univariate Cox regression analysis (A)and multivariate Cox regression analysis (B) of the vitamin D-related signature and clinical factors with overall survival.

## Clinical correlation analysis

A correlation analysis was carried out between our prognostic model and patients' clinical parameters in all 393 CRC cases. With the gradual increase of clinical stage, T and N classification, the risk score of the prognostic model was getting higher (Figs. 6A–6C), suggesting that the signature was reliable. However, it was not shown that the M classification was associated with clinical traits (Fig. 6D).

## Establishment of the nomogram model

To predict the survival probability of CRC patients in 1, 3, and 5 years, a nomogram model was established. Calculating the score of clinical factors and vitamin D-related gene signature, a straight line was generated to evaluate the probability of 1-,3- and 5-year survival at each time point in TCGA dataset. According to this nomogram model, patients in the low-risk group showed a better survival probability. Furthermore, the calibration curve for predicting patient survival presented that the rate of predicted 1-,3- and 5-year survival closely paralleled the actually observed ratio of the training set, respectively

**Table 2 The characteristics of patients in the TCGA database.**

| Parameter | Whole cohort (n = 393) | Low risk (n = 200) | High risk (n = 193) | P-value |
|---|---|---|---|---|
| **Age** | | | | |
| <70 | 178(45.3%) | 100(50%) | 109(56.5%) | 0.2359 |
| >=70 | 215(54.7%) | 100(50%) | 84(43.5%) | |
| **Gender** | | | | |
| Male | 207(52.7%) | 101(50.5%) | 106(54.9%) | 0.4373 |
| Female | 186(47.3%) | 99(49.5%) | 87(45.1%) | |
| **Clinical stage** | | | | |
| I | 66(16.8%) | 43(21.5%) | 23(11.9%) | 0.0000476 |
| II | 158(40.2%) | 93(46.5%) | 65(33.7%) | |
| III | 105(26.7%) | 44(22.0%) | 61(31.6%) | |
| IV | 64(16.3%) | 20(10.0%) | 44(22.8%) | |
| **T classification** | | | | |
| T1 | 9(2.3%) | 8(4.0%) | 1(0.5%) | 0.0006668 |
| T2 | 66(16.8%) | 40(20%) | 26(13.5%) | |
| T3 | 272(69.2%) | 139(69.5%) | 133(68.9%) | |
| T4 | 46(11.7%) | 13(6.5%) | 33(17.1%) | |
| **N classification** | | | | |
| N0 | 233(59.3%) | 139(69.5%) | 94(48.7%) | 0.00001556 |
| N1 | 90(22.9%) | 41(20.5%) | 49(25.4%) | |
| N2 | 70(17.8%) | 20(10.0%) | 50(25.9%) | |
| **M classification** | | | | |
| M0 | 329(83.7%) | 180(90%) | 149(77.2%) | 0.0009721 |
| M1 | 64(16.3%) | 20(10.0%) | 44(22.8%) | |

**Table 3 Univariate and multivariate analyses for overall survival.**

| Univariate analysis | | | | Multivariate analysis | | |
|---|---|---|---|---|---|---|
| Variables | HR | HR.(95% CI) | P-value | HR | HR.(95% CI) | P-value |
| **age** | 1.03 | [1.01–1.05] | 0.001519 | 1.05 | [1.03–1.07] | 9.14E−07 |
| **gender** | 1.03 | [0.67–1.57] | 0.891596 | 1.34 | [0.87–2.08] | 0.185489 |
| **stage** | 2.19 | [1.71–2.79] | 3.09E−10 | 1.70 | [0.83–3.49] | 0.14978 |
| **T** | 2.89 | [1.89–4.42] | 9.07E−07 | 1.70 | [1.01–2.88] | 0.046063 |
| **N** | 1.98 | [1.54–2.55] | 9.84E−08 | 1.14 | [0.72–1.78] | 0.583533 |
| **M** | 4.08 | [2.61–6.38] | 7.58E−10 | 1.35 | [0.51–3.58] | 0.540915 |
| **riskScore** | 1.09 | [1.06–1.11] | 3.31E−11 | 1.09 | [1.06–1.12] | 1.18E−08 |

(Figs. 7A, 7B–7D), indicating the agreement between model prediction and reality. Overall, the high accuracy of vitamin D-related gene signature was confirmed by the nomogram model and calibration curve.

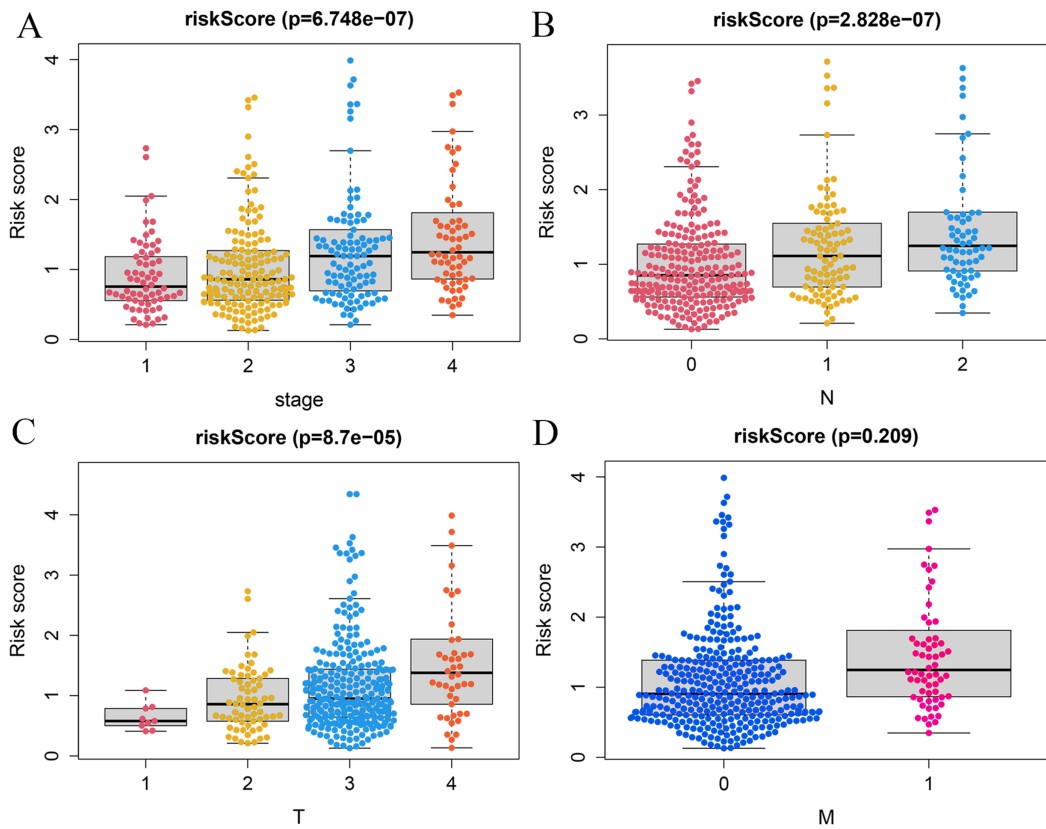

**Figure 6 Correlation analysis of vitamin D-related signature with clinical variables.** Association of risk scores with TNM stages (A), T classification (B), and N classification (C) and M classification (D).

## DISCUSSION

It has been reported that vitamin D is closely related to CRC (*Zhou et al., 2020*). Epidemiology data suggested that vitamin D deficiency is linked to high incidence and/or mortality by CRC (*Ferrer-Mayorga et al., 2019*). Vaughan's research showed that colorectal cancer (CRC) surgical resection decreased circulating vitamin D, whose level is a prognostic biomarker associated with poor survival (*Vaughan-Shaw et al., 2020*). Recent research indicated that vitamin D inhibits cell proliferation and maintains the stem cell phenotype by increasing several stemness-related gene expressions (*Fernández-Barral et al., 2020*). Besides, one of the main vitamin D-related genes, CYP24A1 was found to be negatively associated with the prognosis of prostate cancer (*Khan et al., 2019*). Therefore, vitamin D-related genes are critical in the development and progression of carcinogenesis. Studies have shown that some vitamin D-related genes are of great importance in CRC (*Pálmer et al., 2001*). However, only a few vitamin D-related genes have been studied in depth to confirm their tumor progression role. The functions of most vitamin D-related genes are not well understood, so clarifying the possible role of vitamin D-related genes in prognosis will support the further research in this field. Moreover, the 5-year survival rate of CRC patients is not satisfactory (*McQuade et al., 2017*), so better prognostic models are urgently needed. At present, it is not clear whether vitamin

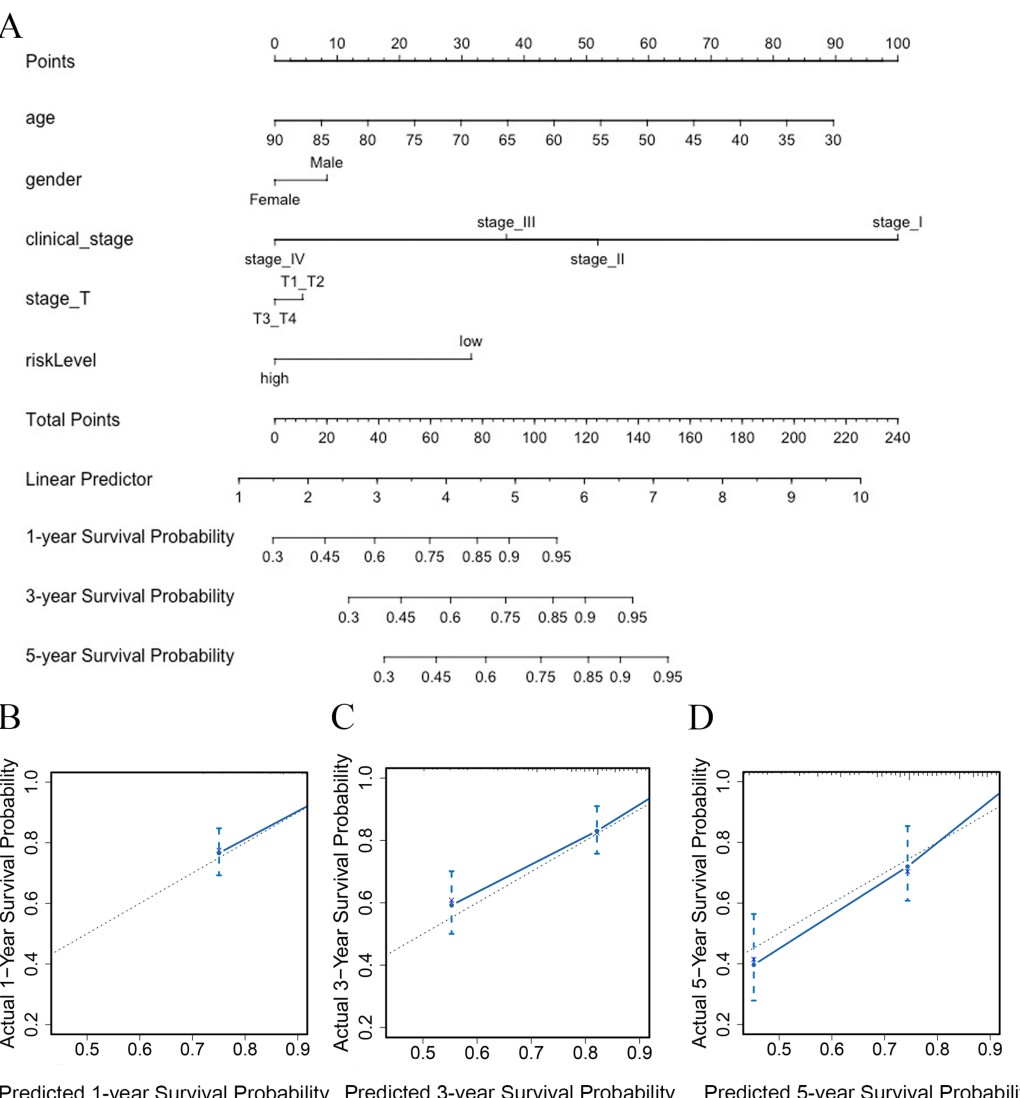

**Figure 7 The nomogram constructed to predict overall survival (OS) in the clinical setting presents good prediction ability.** A nomogram created by the combination of the risk score and TNM stage to predict the OS of CRC (A). Calibration charts predicting 1-, 3- and 5-year survival in TCGA dataset. The horizontal axis and vertical axis represent the predicted survival probability and the actual survival probability (B–D).

D-related genes can predict the prognosis of patients with CRC. Therefore, a vitamin D-related gene model is launched to predict the prognosis of CRC patients and reveal the important function of vitamin D-related genes in the future.

Five RNA sequencing databases were enrolled in the present study, including TCGA, GSE17536, GSE103479, GSE39582, and GSE17537; and clinical characteristics were collected from independent international databases. We identified 17 vitamin D-related genes in CRC from TCGA database. The accuracy of the model was further validated by evaluating the survival analysis, ROC curve, and independent prognostic analysis. The signature consisted of 17 vitamin D-related genes significantly correlated with the

overall survival in CRC patients. The result of Kaplan–Meier analysis supported that patients in the high-risk group had poorer overall survival (OS) than those in the low-risk group. Subsequently, we used a time-ROC analysis to test its performance in the training and validation groups at different time points. The results were comparable compared with other studies.

In this study, we developed a novel vitamin D-related gene signature to predict the prognosis of CRC. Tumors with low risk-score displayed substantially better prognosis in both training and testing sets. Kaplan–Meier survival analysis and ROC analysis were utilized to evaluate the predictive efficiency of vitamin D-related gene signature in the training set and testing sets. Moreover, univariate and multivariate Cox regression analysis revealed that this signature is a firm tool that acts as an independent prognostic factor for OS in CRC patients. Results showed relationships between the signature and some clinicopathological characteristics including the clinical stage and T/M/N stages, but the association was not found with gender. The reason may be that there are no differences in vitamin D-related genes in different genders. Incorporated with independent clinical risk factors, the vitamin D-related gene signature presented good performance. Subsequently, based on the vitamin D-related gene prognostic signature, a nomogram was built to predict 1-, 3-, and 5-year OS. Nomogram integrating clinical factors with Vitamin D-related gene signature predicts the prognosis of CRC patients in clinical practice, providing enlightenment in identifying more gene targets for CRC treatment. The calibration curve also showed agreement between model prediction and reality in the training set.

The signature was composed of 17 vitamin D-related genes with prognostic capability. Among them, fourteen vitamin D-related genes (CYP24A1, TGFB1, IGFBP2, IGFBP3, VGF, AGAP2, DENND6B, LRRC8A, BCL6, FCER2, ELL, CD36, AGPAT1, and TMCO6) were associated with high risk and three (NCOA7, GSR, and MPC1) were identified as protective factors. Researches have demonstrated that dysregulation of vitamin D-related pathways contributes to the pathogenesis of CRC (*Dou et al., 2016*; *Sluyter, Manson & Scragg, 2021*; *Vladimirov et al., 2020*). Studies have demonstrated that some vitamin D-related genes of this signature are involved in metabolism, including AGPAT1, VGF, and MPC1 (*Agarwal et al., 2017*; *Stephens et al., 2017*; *You, Lee & Roh, 2021*). As we know, metabolism reprogramming is one of the important characteristics in tumorigenesis (*Pavlova & Thompson, 2016*); considering their biological function in the regulation of metabolism, it is reasonable to speculate that these genes may be involved in metabolic reprogramming in CRC. In head and neck cancer, MPC1 regulates ferroptosis in vivo and in vitro (*You, Lee & Roh, 2021*); meanwhile, recent studies have confirmed the significant role of ferroptosis in CRC (*Xu et al., 2021*); so, it may be promising to further study whether MPC1 can modulate ferroptosis in CRC. Besides, AGPAT1 deficiency mice have reduced plasma glucose levels and body weight and alteration in lipid synthesis (*Agarwal et al., 2017*). VGF gene expression was lower in obese patients, indicating that VGF may be significantly correlated with obesity (*Koc et al., 2021*); It has been reported that VGF is associated with energy balance and glucose homeostasis (*Ferri et al., 2011*; *Stephens et al., 2017*).

According to our findings, CYP24A1, IGFBP2, and IGFBP3 were related to high risk in CRC. Previous studies also indicated that these genes were identified as important oncogenes in human cancers, including CRC (*Ben-Shmuel et al., 2013*), pancreatic cancer (*Kendrick et al., 2014*), breast cancer (*Dean et al., 2014*), hepatocellular carcinoma (HCC), gastroesophageal cancer, and so on (*Jin et al., 2020*). It was confirmed that the abnormal expression of CYP24A1 was related to cancer risk and might contribute to tumor aggressiveness (*Hu et al., 2019*; *King et al., 2010*). Growing evidence indicates that IGFBP2 plays an essential role in several key oncogenic processes, such as tumor cellular proliferation, epithelial to mesenchymal transition, stemness, invasion, angiogenesis, immunoregulation, and migration (*Li et al., 2020*). In our signature, CD36, ELL, and MPC1 are important genes associated with CRC prognosis. This finding is consistent with the following experimental results. Rainer Hubmann et al's study revealed that NOTCH3 and CD36 influence the uptake, tissue distribution, and activation of vitamin D (*Kiourtzidis et al., 2020*); inhibition of CD36 partially reversed the migration promotion effect of CAFs on CRC cells. By reducing CD36 level in vivo, the migration ability of CRC cells is significantly repressed (*Fedirko et al., 2019*). ELL has been identified as a potential tumor suppressor by interacting with c-Myc and suppresses colon tumor xenograft growth (*Chen et al., 2016*). It was found that MPC1 was downregulated in CRC and its low expression was correlated with poor prognosis; Mechanically, decreased MPC1 enhanced tumor metastasis by activating the Wnt/β-catenin pathway (*Wang et al., 2021*). In addition, the function of few genes might not be consistent with our findings. For example, LRRC8A was identified to regress the proliferation of the CRC cells (*Fujii et al., 2018*; *Xu, Wang & Shi, 2020*). While LRRC8A is a high-risk factor in our model, so further studies are expected to explore its function in CRC. The role of AGAP2, BCL6, GSR, and FCER2 in CRC is not well explored, but they have been studied in other diseases. AGAP2 is closely related to TGFβ1. In activated hepatic stellate cells (HSC), silencing AGAP2 expression diminished proliferation and migration in response to TGFβ1 and led to pro-fibrotic effects (*Navarro-Corcuera et al., 2020*). Glutathione reductase (GSR) is a key regulator in disease; GSR-null mice were susceptible to HCC induced by chemicals and the liver had higher DNA damage markers (*McLoughlin et al., 2019*), which may suggest its protective role in cancer. GSR-KO mice also implicated the GSH system as the main regulator of lung development (*Robbins et al., 2021*). The main research of BCL6 is focused on lymphoma and it was considered as a therapeutic target (*Leeman-Neill & Bhagat, 2018*). FCER2 (CD23) surface expression is mutually exclusive in chronic lymphocytic leukemia (CLL) (*Hubmann et al., 2020*). Moreover, the role of DENND6B, NCOA7 and TMCO6 in cancer is largely unknown, our results indicated their important role in colorectal cancer; further studies are expected to reveal their biological function.

To date, there is no published studies using vitamin D-related gene signatures to predict prognosis of CRC. Our study is the first one to establish a vitamin D-related signature for prediction of CRC prognosis. The external verification can increase the credibility of this model, so our research was verified by four external databases (GSE17536, GSE103479,

GSE39582, and GSE17537) to further prove its accuracy, which making our results more convincing. Moreover, a large amount of CRC patients enrolled, our research was conducted with a reasonable sample size and achieved ideal performance, making the future applications more extensive. Our research confirmed that vitamin D and vitamin D-related genes play a vital role in CRC and supports that vitamin D-related genes may be potential therapeutic targets. Compared to traditional clinical risk scoring, incorporating our vitamin D-related gene signature with clinical risk factors would benefit prognostic prediction. It may provide a theoretical basis for predicting whether vitamin D supplementation is effective in CRC patients. For the first time, our research showed that vitamin D-related genes are associated with CRC prognosis. It strongly supports the vital function of vitamin D in tumors, suggesting that further studies of the mechanisms of vitamin D and vitamin D-related genes in tumors are needed.

However, we have to mention the following defects in the present study. First of all, the validation of vitamin D-related genes in CRC clinical samples will make our conclusion more reliable. Meanwhile, basic experiments are critical to revealing the mechanism of vitamin D-related genes in CRC.

## CONCLUSION

In summary, we first identified and constructed a vitamin D-related gene signature to predict the prognostic outcome of CRC patients, which was an independent prognostic marker for overall survival. Our current study highlighted that vitamin D-related genes played an essential role in the prognosis of CRC and deepened the understanding of vitamin D-related genes. Efforts to reveal the role of vitamin D in CRC will help develop more rational treatment strategies in the future. These findings would help provide new therapeutic targets and prognostic markers for CRC.

### Funding

This study was supported by the National Natural Science Foundation of China (Qiu Zhao, No. 81870390), and the Zhongnan Hospital of Wuhan University Science, Technology and Innovation Seed Fund (Lan Liu, No. CXPY2017033). The funders had no role in study design, data collection and analysis, decision to publish, or preparation of the manuscript.

### Grant Disclosures

The following grant information was disclosed by the authors:
National Natural Science Foundation of China: 81870390.
Zhongnan Hospital of Wuhan University Science, Technology and Innovation Seed Fund: CXPY2017033.

### Competing Interests

The authors declare that they have no competing interests.

## Author Contributions

- Luping Bu conceived and designed the experiments, performed the experiments, analyzed the data, prepared figures and/or tables, authored or reviewed drafts of the paper, and approved the final draft.
- Fengxing Huang performed the experiments, analyzed the data, authored or reviewed drafts of the paper, and approved the final draft.
- Mengting Li performed the experiments, prepared figures and/or tables, and approved the final draft.
- Yanan Peng analyzed the data, prepared figures and/or tables, and approved the final draft.
- Haizhou Wang analyzed the data, authored or reviewed drafts of the paper, and approved the final draft.
- Meng Zhang analyzed the data, prepared figures and/or tables, authored or reviewed drafts of the paper, and approved the final draft.
- Liqun Peng analyzed the data, prepared figures and/or tables, authored or reviewed drafts of the paper, and approved the final draft.
- Lan Liu conceived and designed the experiments, authored or reviewed drafts of the paper, and approved the final draft.
- Qiu Zhao conceived and designed the experiments, authored or reviewed drafts of the paper, and approved the final draft.

## Data Availability

Data is available at TCGA-COAD and NCBI GEO: GSE17536, GSE103479, GSE39582, and GSE17537.

https://portal.gdc.cancer.gov/projects/TCGA-COAD.

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
