# Peer review of "Identification of Vitamin D-related gene signature to predict colorectal cancer prognosis"

_PeerJ, doi:10.7717/peerj.11430_

## Round 0.1 · original submission · Major Revisions

Dear Dr. Bu and Dr. Huang,

Your manuscript has been reviewed by three experts in the field and two reviewers are in agreement that analysis of the results is not adequately conveyed in the manuscript. Please carefully read the reviews especially reviewers 2 and 3 and modify the manuscript accordingly.

·

Basic reporting

The manuscript by Bu and Huang et al. entitled "Identification of Vitamin D-related gene signature to predict colorectal cancer prognosis" constructs a vitamin D-related gene signature to predict prognosis in Colorectal cancer (CRC), which is one of the most common malignant carcinomas worldwide with poor prognosis. Vitamin D and vitamin D-related genes play a vital role in CRC.

Experimental design

Very well wrote.

Validity of the findings

TCGA data validate with GEO set

Additional comments

The authors wrote this manuscript very well.

Need to highlights D-related signature gene in discussion and wrote previous studies related to those in CRC or other cancers in detail.

·

Basic reporting

No comment

Experimental design

No comment

Validity of the findings

No comment

Additional comments

The concerns are in the attachment.

·

Basic reporting

The manuscript ' Identification of Vitamin D-related gene signature to predict colorectal cancer prognosis’ by Bu et al using a Vitamin D related gene signature model to predict the prognosis in colorectal cancer. However, there are some concerns which the authors need to address in this manuscript.
The authors need to improve their language used in this manuscript, as it is very simple in many places rather than being scientific.

Experimental design

The authors have clearly indicated that they have obtained the clinical information of CRC patients. There is no proper information regarding the source and there is no statement on whether this study was reviewed by an institutional ethics committee or board.

Validity of the findings

The authors should describe more in detail about the univariate cox analysis performed in the CRC samples and present those results. It is not clear in the results how those 17 genes were selected. Also, it is not clear as to how and on what basis those 17 genes were classified further as high and low risk genes.

The construction and validation of this gene analysis has been based on the ‘hazard ratio’, but it has not been made clear anywhere in this manuscript as to what this means and how this is calculated. This is very important because this is the foundation for this model.

Under the high risk and low risk genes classification, the authors have 14 and 3 genes respectively. The authors should explain the biological importance of both these genes and provide more evidence as to why they are important.

The authors description of the univariate and multivariate analysis in table 3 is not clear as stated in the text of the manuscript. Also, it is not clear as to what motivated the authors to perform this analysis.

---

## Round 0.2 · Minor Revisions

Dear Dr. Bu and Dr. Huang,

Although the two reviewers have formally accepted the manuscript, I wanted to give minor revisions because I felt the comments from reviewer 3 are important and need to be addressed. Please revise and resubmit the manuscript.

·

Basic reporting

The authors fulfilled the comments and I do not have further concern. The revised manuscript looks great and acceptable, after other reviewer's comments.

Experimental design

The authors provided details in a revised version of the article.

Validity of the findings

They also validate their data in more than one database.

Additional comments

Thanks for providing responses.

·

Basic reporting

The quality of this article has improved considerably post-revision. Figures also have a better resolution.

Experimental design

The authors have corrected concerns about experimental design and information provided.

Validity of the findings

The results have now been validated by adding more datasets thus making the conclusions more reliable and less-extrapolated.

·

Basic reporting

The manuscript “Identification of Vitamin D-related gene signature to predict colorectal cancer prognosis” by Bu et al using a Vitamin D-related gene signature model to predict the prognosis in colorectal cancer. However, there are some minor concerns which the authors still need to address in the revised manuscript.

Experimental design

N/A

Validity of the findings

1. Appreciate the authors on providing the formula for the hazard ratio. Can the authors validate this formula as to how or on what basis these formula was formulated?

2. Under the high risk and low risk genes classification, the authors have 14 and 3 genes respectively. The authors should explain the biological importance of both these set of genes and provide more evidence as to why they are important.
The response provided by authors for this comment is not satisficatory.
The main reason to ask this question was to understand the context of these genes in terms of their functions in other diseases, so that, it will aide in the analysis in the context of CRC. However, the authors fail to discuss this point clearly again.

---

## Round 0.3 · accepted · Accept

Dear Dr. Bu and Dr. Huang,

I have now accepted the manuscript for which you had submitted a revised version.